# Crispness, the Key for the Palatability of “Kakinotane”: A Sensory Study with Onomatopoeic Words

**DOI:** 10.3390/foods10081724

**Published:** 2021-07-26

**Authors:** Atsuhiro Saita, Kosuke Yamamoto, Alexander Raevskiy, Ryo Takei, Hideaki Washio, Satoshi Shioiri, Nobuyuki Sakai

**Affiliations:** 1Department of Psychology, Tohoku University, 27-1 Kawauchi, Aoba-ku, Sendai 980-8577, Japan; atsuhiro.saita.q5@dc.tohoku.ac.jp; 2Research Institute of Electrical Communication, Tohoku University, 2-1-1 Katahira, Aoba-ku, Sendai 980-8577, Japan; ko.yama@tohoku.ac.jp (K.Y.); shioiri@riec.tohoku.ac.jp (S.S.); 3Division for Establishment of Frontier Sciences of the Organization for Advanced Studies, Tohoku University, 2-1-1 Katahira, Aoba-ku, Sendai 980-8577, Japan; 4Advanced Institute for Yotta Informatics, Tohoku University, 2-1-1 Katahira, Aoba-ku, Sendai 980-8577, Japan; 5Faculty of Psychology, Lomonosov Moscow State University, 11-9 Mohovaya Str., 125009 Moscow, Russia; a.raevskiy@gmail.com; 6Japan Society for Promotion of Science (JSPS), 5-3-1 Kojimachi, Chiyoda-ku, Tokyo 102-0083, Japan; 7Rice Research Institute, Rice Research Institute, Kameda Seika CO., LTD., 3-1-1 Kameda-kogyodanchi, Konan-ku, Niigata 950-0198, Japan; r_takei@sk.kameda.co.jp (R.T.); hideaki_washio@sk.kameda.co.jp (H.W.)

**Keywords:** crispness, mimetic words, food palatability, temporal dominance of sensations (TDS), temporal drivers of liking (TDL)

## Abstract

Crispness is among the most important food textures that contribute significantly to palatability. This study investigated the association between the perceived crispness and palatability of five types of Japanese rice crackers known as “kakinotane.” Two experiments were conducted using the temporal dominance of sensations (TDS) and temporal drivers of liking (TDL) methods. As descriptors for the TDS evaluation, we used 10 Japanese onomatopoeias to indicate various attributes of crispness. We also measured the mastication sounds and electromyography (EMG) activity during mastication. Principal component analysis data revealed that principal component 1, representing moisture characteristics, contributed more than 60% in both experiments. The palatability of the stimulus, which was described as having a very soft, moist, and sticky texture, BETA-BETA, was significantly lower than the others. However, there was no significant relationship between the amplitude of mastication sound or EMG activity and palatability. We demonstrated that naïve university students can discriminate the fine nuances of the crispness of “kakinotane” using the TDS and TDL methods. Our findings also suggested that the onomatopoeias used as descriptors in the TDS method had a greater influence on describing the nuances of food texture than the physiological data.

## 1. Introduction

Crispness is an important factor in food perception and evaluation and plays a key role in food palatability [1]. Many previous studies have investigated the characteristics of food crispness using physical/material or sensory science approaches. It has been suggested that perceived crispness is affected by the hardness of food and the magnitude of force required to crush it [2]. Katz and Labuza revealed that increased water activity of snacks led to a decrease in the crispness evaluation [3]. This effect was confirmed by Seymour and Hamann who reported that an increased moisture level decreased the sensory hardness of snacks [4]. Since crispness is an indicator of freshness and wholesomeness [5], the moisture level in snacks is an indicator of staleness. As crispness is one of the most desirable textual characteristics, we generally prefer crispy snacks and cereals rather than soft snacks [6]. Recent studies have also suggested that acoustic parameters related to food sensory properties, such as crispiness and crunchiness, are positively correlated with satisfaction and pleasantness [7].

While the contribution of the physical properties of food to its crispness has been examined, few studies have examined the perception of crispness. Specifically, Japanese people prefer crispy foods and are more sensitive to different degrees of crispness than North Americans [8]. This study aimed to clarify the role of crispness perception in the palatability of “kakinotane” (Japanese spicy rice crackers), a popular snack in Japan.

A useful methodology for investigating the sensory characteristics of foods is the temporal dominance of sensations (TDS) method [9], which evaluates dominant sensations and their temporal dynamics using descriptors from a proposed sensory attribute list. The TDS method has been used to accurately assess the perception of food texture, odor, taste [10], and textural complexity in model foods [11]. In this study, we used the TDS method based on mimetic words or onomatopoeias.

Mimetic words are an example of the sound symbolic aspect of language, where the acoustic representation of the word has a connection with its meaning. Moreover, mimetic words imitate sounds or other sensory experiences, rather than describe higher cognitive concepts. As texture is a sensory property, it can be described more effectively by sensory words such as onomatopoeias, rather than by semantic words [12]. Previously, onomatopoeias in Spanish, French, and German languages have been used for expressing the eating sensation of fruits and vegetables, and these words can sufficiently describe food perception [13]. Hayakawa et al. showed that onomatopoeias are also frequently used for expressing food textures in the Japanese language [14].

Approximately half of the 135 Japanese onomatopoeic words for texture have a recognition rate of at least 75%, and 66 out of these words were correctly recognized by over 90% of the nonprofessional population [15]. As there are many such words in the Japanese language, using them to describe various aspects of food sensations may theoretically help solve the problem of describing crispness according to sensory attributes [16,17].

Another important reason for using onomatopoeic words in this study is their connection with the visual cortex area in the brain, whereby experiences are felt just by reading or hearing the word [18]. Because of the multisensory nature of such terms, they may be used as descriptors for the crispness and for expressing tiny nuances, thus, providing associations with perceived palatability. A recent study indicated that onomatopoeic expressions influenced consumer expectations of the product [19] and can be used in the food industry. Thus, Japanese onomatopoeic words for crispness provide an effective evaluation mechanism for the current research.

Another important method used in this study is the temporal drivers of liking (TDL) method, developed by Thomas et al. [20]; a TDS session and a liking session are used for participants to continuously evaluate the palatability of a presented stimulus. The types of sensory attributes used in the TDS session that contribute the most to palatability are explored during the liking session. In this study, we investigated which textural characteristic was essential for food palatability using the TDL method.

Different psychological factors may impact our evaluation of foods. Interestingly, it has been suggested [21] that vocabulary plays an important role. This study aimed to determine the associations between onomatopoeic expressions and crispness/palatability of “kakinotane” rice crackers using the TDS and TDL methods. Japanese participants were asked to choose onomatopoeias that describe the crispness of five kinds of “kakinotane” and to evaluate the palatability of the same samples.

## 2. Experiment 1

### 2.1. Materials and Methods

#### 2.1.1. Participants

Twenty-one university students (eight women and 13 men; mean age, 20.5 ± 3.2 years) participated in this study after providing written informed consent. All participants were native Japanese speakers. One participant was excluded from the analysis because of missing data.

The sample size of the study was determined with the reference of those in the preceding studies [10,22]. All the participants were students, graduate students, and researchers of our department, and had enough experience in experimental psychology of sensation and/or sensory studies of foods. The experimental protocols were approved by the ethics committee of Tohoku University (No. 2019-008).

#### 2.1.2. Stimuli and Apparatus

Five types of “kakinotane” were presented as samples labeled with the name S, N, K, Kr, and Kc. Three of the samples (S, N, and K) were produced by representative brands of “kakinotane” and chosen based on varieties of textural characteristics by the sensory panel of the Japanese manufacturer. These samples were commercially available products produced by different manufactures (Figure 1). Sample Kr was a moisturized and re-dried version of K, and Kc was a crushed version of K. The latest was used to investigate the difference in palatability between normal and de-structured foods, as the de-structured foods were suggested not to have the texture of the first bite. The order of the samples was counterbalanced using a randomized block design.

A pilot study was performed to select the most appropriate onomatopoeic words for describing the crispness of “kakinotane.” Ten university students and researchers wrote down the onomatopoeias describing the crispness of each sample while eating it for 90 s.

The 10 most frequently used words were selected and used as descriptors in the TDS session: BARI-BARI, BORI-BORI, GARI-GARI, ZAKU-ZAKU (for hard textures), PARI-PARI, PORI-PORI, KARI-KARI (for more crumbly textures), SAKU-SAKU, SARA-SARA (for light crispy textures), and BETA-BETA (wet and sticky texture). For a detailed explanation of the terms see Appendix A.

The experiment was conducted using the sensory analysis software MagicSense 3.1.6 (Taste Technology, Inc., Tokyo, Japan) for Windows. The participants responded using a touchscreen-based personal computer. Responses were recorded at a sampling rate of 5 Hz with MagicSense.

#### 2.1.3. Procedure

The experiment consisted of the TDS session and the liking session. Both sessions were conducted on the same day. The order was counterbalanced: half of the participants started with the TDS session and the other half with the liking session.

In the TDS session, the participants were asked to keep chewing a sample for 60 s and evaluate the crispness using the given descriptors. Four types of samples were presented in paper cups containing five pieces of each product (approximately 2 g). Within 5 s of placing a sample in their mouth, the participants were asked to touch the button on the screen to start the trial. While masticating at a free pace, the participants were requested to select the most appropriate descriptor of the crispness from the 10 descriptors displayed on the screen. When their perception of crispness changed, they were asked to select another descriptor. The participants were asked not to choose any of the descriptors if they had already swallowed the sample or if none of the words described the perception. After each trial, a 1-min rest period was provided, during which, the participants rinsed their mouths with water.

To measure the palatability of the samples, the liking session was conducted in accordance with the TDL method [20]. In the liking session, the task was to successively evaluate the palatability of the samples. The participants were asked to begin masticating the sample within 5 s of placing the sample into their mouths and to evaluate the palatability using the linear scale (“1” very unpalatable to “9” very palatable) displayed on the touchscreen personal computer running MagicSense. They were asked to change the rating when the palatability changed. The trial was completed when the participants no longer perceived any crispness or after the sample was swallowed. The next trial was initiated after the participants had rinsed their mouths during the 1-min rest.

#### 2.1.4. Data Analyses

The TDS data were analyzed using the MagicSense software to obtain the TDS band plots of the perceived crispness. Additionally, for the duration of each descriptor chosen for each sample in the TDS session, principal component analysis (PCA) was performed. One-way analysis of variance (ANOVA) was performed for the mean liking values over time in each trial. A post-hoc comparison was performed using Ryan’s method when a significant main effect was observed.

The data obtained from the first 30 s of the TDS and liking sessions were combined and analyzed using the TDL method with Python 3.7 to calculate the liking while dominant (LWD) scores for each sample, according to Thomas et al. [20]. The data from the second half of the data were excluded, as most of the participants had completed the evaluation within the first 30 s and had no data after 30 s. The LWD score is a weighted-liking score for each attribute used in the TDS session. These individual LWD scores were centered (CLWD), and each CLWD score was tested for equality to the theoretical mean of 0 using a one-sample *t*-test. When a CLWD score was significantly (α = 5%) higher or lower than 0, it indicated a positive or a negative driver of liking, respectively [23].

### 2.2. Results

The TDS band plots for each sample are depicted in Figure 2, demonstrating the sequence of the dominant attributes in the same product [23]. Sample Kr was mainly described using BETA-BETA in the second half of the session. Regarding the remaining samples, the term KARI-KARI was selected most often in the first half of the session, followed by SARA-SARA and BETA-BETA in the second half. Moreover, BORI-BORI was used to describe the majority of the samples.

PCA on attribute durations revealed that the two principal components (PCs) with variance resulted in a contribution rate of 84% (Figure 3). The variance of PC1 and PC2 contributed to 52% and 32%, respectively. These results suggested that the crispness of “kakinotane” should be evaluated using two scales. The PC1 axis represents moisture characteristics (expressed by BETA-BETA–SAKU-SAKU), whereas the PC2 axis represents hardness (expressed by GARI-GARI–SAKU-SAKU). A relatively high contribution rate of PC1 theoretically indicated that the crispness of “kakinotane” was mainly evaluated according to the moisture characteristics.

Figure 4a shows the time-course of changes in liking scores for all the samples. A one-way repeated measures ANOVA was performed for the average liking scores averaged over time and indicated a significant main effect of the food sample [*F*(4, 76) = 31.84, *p* < 0.05, *η_p_^2^* = 0.63] (Figure 4b). A post-hoc multiple comparison test revealed that the liking score for Kr was significantly lower than that for the remaining four samples (*p*s < 0.05 with Ryan’s test) and that the liking score for N was significantly lower than that for S (*p* < 0.05).

The CLWD scores for the five samples are depicted in Table 1. The onomatopoeia KARI-KARI had a positive CLWD score for samples S, Kc, and N. However, PORI-PORI had a negative CLWD score for samples S and N, and ZAKU-ZAKU had a negative CLWD score for samples K and Kc. However, these CLWD scores failed to reach significance (*p*s > 0.05) because of the small samples (i.e., six, five, six, six, and eight values for each sample).

### 2.3. Discussion

In the current experiment we measured the perceived crispness of five samples of “kakinotane” As previously confirmed by the TDS data of the preceding studies, hardness and crispness are dominant sensations at the beginning of mastication, but stickiness becomes more dominant at the end of the mastication period [24].

Based on the obtained TDS data, PCA was performed. Our findings suggested that two PCs are mostly responsible for the crispness sensation: PC1 is the moisture characteristic of the sample, and PC2 is the hardness, with a high predominance of PC1. As moisture, expressed by BETA-BETA, is the main factor in crispness, the CLWD score revealed that palatability could be expressed negatively using only one descriptor. The remaining descriptors were not effective reflections of palatability.

Consistent with previous findings [3,4], moisture content played a significant role in crispness. It has also been suggested that verbal expression has a certain impact on palatability; especially, some words reflect a negative image of taste. In agreement with previous observations [25], voiced consonants articulated by closing the mouth (for example, [b]) have unpleasant associations with taste and texture: the results of Experiment 1 concerning the low liking results for BETA-BETA display the same tendency. Moreover, clear linguistic sounds (like [s]), as in SAKU-SAKU, provide a more pleasant image. This trend in verbal expression of palatability also correlates with the moisture level.

Experiment 1 had inherent methodological flaws. First, the duration of the trials in the TDS and liking sessions were different because the liking sessions were completed when the participants no longer perceived crispness or after swallowing the sample. In contrast, the duration of the TDS sessions was fixed at 60 s. The CLWD is the mean liking score weighted by the dominant duration of each descriptor. In the calculation, the durations of the sessions were averaged. Therefore, differences in the duration of both sessions might have affected the results. To solve this issue, in the next experiment, we investigated the temporal dynamics of crispness evaluation in a shorter time window of 30 s. We set the pace of mastication to 1 Hz to control the speed with which crispness was changing.

Second, in Experiment 1, we considered only the subjective evaluation of crispness, and we did not investigate the physical and physiological properties of the crispness of “kakinotane.” As suggested by previous studies [26,27], measuring mastication sounds and EMG activity during mastication could help improve our understanding of the physical properties of foods. According to Vickers, variations in the perceived crispness could be sufficiently identified by acoustic and force-deformation measurements [27,28]. As we used onomatopoeias as the descriptors of crispness, a correlation between the verbal expressions and physical properties of the samples was expected. Therefore, in Experiment 2 we measured and analyzed mastication sounds and EMG activity during mastication of each sample.

## 3. Experiment 2

### 3.1. Materials and Methods

#### 3.1.1. Participants

Twenty university students (12 women and eight men; mean age, 20.7 ± 1.2 years) participated in the study after providing written informed consent. All the participants were native Japanese speakers, untrained on TDS. The participants did not include those from Experiment 1. The experimental protocols were approved by the ethics committee of Tohoku University (No. 2019-008).

#### 3.1.2. Stimuli and Apparatus

The food samples were identical to those used in Experiment 1. The experimental procedure was conducted with the same program and a personal computer. Mastication sounds were recorded using a handheld audio recorder (DR-07X, TASCAM, Montebello, CA, USA) with a windproof bore close to the mouth and binaural microphones (CS-10EM, Roland Corp., Shizuoka, Japan) inserted into the ears (Figure 5). Sound signals were processed through a microphone amplifier (AT-MA2, Audio-Technica, Tokyo, Japan) and a USB audio interface (Fireface UCX, RME, Haimhausen, Germany). The activity of the masseter muscles during mastication was measured using EMG with a wireless biomedical sensor (Biosignalsplux, Plux, Arruda dos Vinhos, Portugal) at a 4000-Hz sampling rate. EMG activities were recorded from the left and right masseters using bipolar surface electrodes (Figure 5) according to Endo et al. [29,30].

#### 3.1.3. Procedure

The experiment consisted of three sessions: the TDS, liking, and sound/EMG recording sessions. All the sessions were conducted in the above order in 1 day. In the TDS and liking sessions, the procedures were identical to Experiment 1 except for the pace of mastication and the duration of each trial. The participants were asked to masticate at 1 Hz in time with metronome clicks. The duration of each trial and the rest period were 30 s.

In the sound/EMG recording session, the participants placed the binaural microphones into their ears and masticated the samples while keeping their mouths close to the audio recorder. Additionally, the bipolar electrodes for EMG recording were attached to the masseter muscles. The participants touched the button on the screen to begin the trial. After placing the sample into their mouths for 3 s, they were asked to masticate the sample for 30 s at 1 Hz monitored by a timer displayed on the computer screen. They began the next trial after rinsing their mouths during a 30-s rest period.

#### 3.1.4. Data Analyses

The TDS and liking data were analyzed as in Experiment 1. Power spectrum analysis of the mastication sound data was performed using only the first 5 s of each trial as the amplitude of sounds decreased over time (Figure 6). The amplitude spectrum was calculated for each sample, and a one-way repeated measures ANOVA was performed using the mean amplitude spectrum across the frequency components of the samples. Only the data of the frequency components <1000 Hz were analyzed, as notable fluctuations in the amplitude of the components >1000 Hz were not observed. Data from four participants were excluded from the analysis because of recording failure.

EMG data for each sample were analyzed using MATLAB and Signal Processing Toolbox (Mathworks, Inc., Natick, MA, USA). Data from the electrodes on the left and right masseter muscles were averaged and converted to the averaged rectified value by smoothing using a 5-Hz low-pass filter. Data from 10 participants were excluded from the EMG analysis because of a failure in collecting all data, as the electrodes were removed from some of the participants.

### 3.2. Results

TDS band plots for each of the five samples are depicted in Figure 7. For Kr, BETA-BETA was almost the only selected word. For the remaining samples, the descriptors KARI-KARI and BARI-BARI were selected most often in the beginning, later changing to SAKU-SAKU, SARA-SARA, and BETA-BETA.

PCA of the TDS data indicated that two PCs that had variance combined to a contribution rate of 98%. PC1 (86%) was the moisture characteristics of the sample (expressed by BETA-BETA–SAKU-SAKU), and PC2 (12%) was the feeling of the easiness of mastication (expressed by BARI-BARI–ZAKU-ZAKU) (Figure 8).

As in Experiment 1, liking scores for all the samples decreased with time (Figure 9a). A one-way repeated measures ANOVA was performed for the averaged liking scores over time and revealed a significant main effect of the sample [*F*(4, 76) = 72.04, *p* < 0.05, *η_p_^2^* = 0.79] (Figure 9b). A post-hoc multiple comparison test (Tukey’s test) revealed that the liking score for Kr was significantly lower than that for the remaining four samples (*p*s < 0.05), and the liking score for N was significantly lower than that of S (*p* < 0.05).

The CLWD scores for all five samples are presented in Table 2. As in Experiment 1, BETA-BETA and SARA-SARA were significantly negative TDLs. BARI-BARI and BORI-BORI were significantly positive TDLs, which contradicts the results of Experiment 1.

Figure 10 shows the results of the power spectrum analysis of mastication sounds. All samples had similar spectra peaks of approximately 50 and 900 Hz. A one-way repeated measures ANOVA revealed a significant main effect of the food sample [*F*(4, 60) = 2.54, *p* < 0.05, *η_p_^2^* = 0.15]. A post-hoc multiple comparison test (Tukey’s test) revealed that the amplitude spectrum of Kc was significantly higher than that of Kr (*p*s < 0.05). The mean amplitude spectrum of the mastication sounds was positively correlated with the mean liking results (Figure 11a; *r* = 0.63, *p* = 0.08).

Examples of EMG signals from a single participant are presented in Figure 12a. Figure 12b depicts the mean peak voltage of EMG activities for each sample. A one-way repeated measures ANOVA revealed no significant main effect of the food sample [*F*(4, 36) = 0.81, *p* = 0.53, *η_p_^2^* = 0.083]. The mean peak voltage of the EMG was positively correlated with the mean liking results (Figure 11b; *r* = 0.65, *p* = 0.12).

### 3.3. Discussion

The liking score in Experiment 2 was similar to that in Experiment 1. Thus, it appears that mastication speed and ingestion time do not affect the evaluation of palatability.

The TDS band plot indicated that the participants mostly chose BETA-BETA from the beginning to the end in evaluating sample Kr. It was assumed that they used this descriptor in two meanings. At the beginning of the evaluation, they used it to indicate stickiness of the product surface, and later, they used it to indicate moisture.

The PCA results of the TDS data indicated that the participants evaluated crispness according to the moisture characteristics in Experiment 2 as in Experiment 1. However, PC2 in Experiment 2 was not the hardness of the sample but rather the perception of the size of pieces during mastication. This result likely illustrates the effect of the time window difference between the two experiments. As the duration of the TDS session in Experiment 2 was only 30 s and all the samples remained hard during all trials, the participants could focus more on the size differences of the masticated samples. However, in Experiment 1, the participants mostly evaluated the hard-soft properties during the 60-s time frame, as the hardness decreased with mastication because of saturation with saliva.

The difference between the results of Experiments 1 and 2 could be attributed to the metronome usage, which may have influenced participants’ attention during evaluation. However, previous experimental studies have indicated that the number of chews and the time from ingestion to the first swallow did not significantly change the crispness level [31]; thus, reliable results may be acquired without controlling the chewing pace.

Several previous studies have demonstrated that jaw movements [26,32] and mastication sounds [27,33] differed according to food texture. More recent studies have also indicated that acoustic parameters are highly correlated with the sensory crispness [34,35]. In the present study, the intensity of the EMG activities did not differ between the food samples or the mastication strokes. The EMG activities of samples Kr and N had marginally lower peak voltages and were positively correlated with the liking scores. Moreover, Kr had a significantly lower power spectrum than the remaining samples, and the mastication sounds were positively correlated with the liking score.

However, this study failed to detect other significant effects of mastication sounds and the EMG activities of the masseter muscles on crispness. Inconsistencies between the present and previous studies are likely a result of differences in the mastication action, whereby, previous studies [22,27,36] have analyzed sounds using a few bites of potato chips, fruits, and vegetables masticated with the front teeth. Moreover, in the present study, “kakinotane” was mostly masticated with the molars. Further researches based on the different stimuli should be conducted to investigate the connection between the mastication sounds and jaw movements during mastication, as well as the physical properties of foods.

From the results of Experiment 2, we can suggest that in the case of “kakinotane,” onomatopoeic expressions were more positively correlated with crispness and palatability than the acoustic measurements or EMG activity of mastication. It is hypothesized that these results were caused by the impact of onomatopoeic words describing crispness that exceeded the effect of mastication sounds or EMG activities of the masseter muscles. Similarly, previous research indicated that onomatopoeias activate visual cortical areas in the brain [37] and create emotional response [38]. In addition, the sense of eating is stimulated when the participants read or hear onomatopoeic words [18]. Thus, in the case of similar gustation sensations, recording onomatopoetic words can be regarded as a more effective way of evaluating crispness and palatability than mastication force and sound.

## 4. General Discussion

This study revealed that naïve Japanese university students discriminated between similar samples of “kakinotane” based on their crispness. Moreover, they evaluated their palatability differently, in consistence with previous experimental studies demonstrating that untrained volunteers could successfully distinguish between the different varieties of apples with scores similar to the trained panel [39]. Thus, the discrimination and evaluation procedure used in the TDS method and its applied form proved to be useful for investigating the perception of foods by naïve Japanese consumers.

This study also revealed that the palatability of “kakinotane” depended mainly on its crispness, especially on the moisture characteristics. Moreover, onomatopoeias, describing the moisture characteristics, were the key descriptors of the crispness of “kakinotane” because PC1 in both experiments was described by the words BETA-BETA and SAKU-SAKU (“very moist”–“not moist”). Consistent with this observation, two other studies have reported that the moisture characteristics of snacks significantly affected their crispness [3,4]. However, PC2 differed between the two experiments and displayed a little effect on describing the crispness of “kakinotane.” While the contribution values of PC1 were relatively high (62% in Experiment 1 and 86% in Experiment 2), those of PC2 were less substantial in both experiments (26% and 12%, respectively). Two possible limitations may have caused this inconsistency. First, most of the participants tended to use many attributes for each sample; therefore, few attributes were characterized by a longer dominant duration (Figure 3a and Figure 7a). This result was similar to the findings of Rodrigues et al. [40], in which the untrained individuals used more attributes for a particular sample than the trained individuals during the TDS sessions. Second, mimetic words express perceptional nuances acquired by our own experience; however, they do not have strict definitions. Thus, words that represent similar sounds, such as PARI-PARI and BARI-BARI, may have been used inaccurately. Japanese speakers mainly use 13 onomatopoeic words to describe the fine nuances of food crispness [16]. However, these words display inconsistent semantic properties [16], and their usage demonstrates a larger variety of meanings compared to those of adjectives and adverbs. Tanaka suggested that food texture terminologies also depend on eating habits and traditional foods [41]. Moreover, Yoshikawa et al. suggested that the ability to use onomatopoeic words or adjectives depended on the familiarity with those words [16]. Therefore, future studies should use onomatopoeias that do not sound similar to test familiarity with the attributes among the participants.

The results of the LWD scores in both experiments revealed that BETA-BETA was a negative factor of crispness. BETA-BETA describes the moisture characteristics and appeared in the latter half of the TDS session in all samples. As “kakinotane” is famous for its crispness, BETA-BETA describes a quality that is opposite to its characteristic crispness. It should be noted that Japanese people like the stickiness of cooked rice and have a large glossary of terms describing it [41]. However, BETA-BETA, which has also been used to describe stickiness, does not describe a palatable texture as it means “too sticky.” In this study, it was used to express an unpalatable consistency.

While common negative factors are easily identified, the results of the LWD scores in the two experiments failed to identify common positive factors. This can likely be explained by the differences in the proposed conditions, such as the trial durations or the fixed mastication pace. “kakinotane” is a rice cracker with a porous structure that readily absorbs moisture. Thus, the number of masticatory strokes and the time required for a sample to be saturated with saliva are critical factors in determining the crispness of “kakinotane.” Therefore, the most reasonable time window for the TDL analysis to evaluate the relationship between palatability and crispness should be investigated.

Another important factor of the current investigation was the acoustic measurements of mastication to elucidate the physical indices of crispness; however, no significant differences were observed. These results contradicted the findings of the review by Vickers, which revealed correlations between acoustic factors and tactile cues of crispness [28] and suggested that acoustic variables, such as sound pressure and intensity, were related to the crispness of potato chips and crackers. Our results revealed that the perception of crispness was based more on complex multisensory processing, including the perception of moisture characteristics, than on the simple combination of acoustic and force-deformation cues. This finding can likely be explained by a significant difference between the previous studies and the current research. While most studies on food textures usually focus on comparing perception aroused by various and different stimuli, we investigated the variance of close sensations, thus, exploring the nuances of crispness using the connection with verbal expressions.

Previous studies [42,43] have demonstrated that jaw movements, not the activity of the masseter muscles, are differentiated and affected by the texture of foods. Thus, the role of jaw movements in processing texture perception should be investigated in future studies. The difference between the current and previous studies might be explained by the stimulus diversity. Since all the samples in the present study were “kakinotane,” various samples, including crispy and non-crispy foods, should be compared to reveal the general sensory mechanism responsible for the perception of crispness and palatability. To support this idea, a previous study also failed to detect masticatory differences between several kinds of peanuts [44].

While no specific differences in sound measurements were observed, the participants could identify and evaluate different crispness sensations based on very similar sensation stimuli. Thus, the perceived difference in crispness, as the perceived palatability, was mainly associated with the presented descriptors; therefore, semantic expressions may broaden our sensation sphere. As it has previously been suggested, words, such as “crispy” and “crunchy,” are not descriptive enough to describe a complex of sensory sensations and properties of food comprising sight, flavor, taste, texture, and sound. Moreover, a more peculiar sensory vocabulary would aid in distinguishing and investigate crispness levels in more detail [45,46].

In this study, the participants successfully evaluated palatability by using certain onomatopoeic words; therefore, these words can be an effective instrument for exploring the influence of vocabulary on perception. It has been suggested that various words are required to describe those nuances because of different textures in Japanese food [12]; as it is a linguistic feature that is not shared with most other languages, the conclusions of the current study could be culturally and linguistically limited. Although the link between taste and certain phonemes has been described [25], further research is needed to investigate the possibilities of Japanese onomatopoeias for increasing perceptional sensitiveness in non-Japanese speakers.

Recently, many studies have demonstrated that perceptual experience is significantly affected by the linguistic background. For example, Winawer et al. revealed that language affects color discrimination [47]. Experiments on visual search tasks [48] revealed that perception greatly depended on the meaning of the stimuli. Miller et al. also proved the impact of language on tactile perception [49]. Our results are consistent with those of preceding investigations and suggest existing correlations between vocabulary and the perceived crispness and palatability of foods.

An additional, important limitation should be mentioned. This study was conducted with Japanese participants and used specific Japanese verbal expressions. To validate our hypothesis, further investigations focusing on additional connections between language and perceptual experience, are required. Thus, a range of consequent experiments is being prepared to explore the understanding of Japanese onomatopoeias to describe food texture by non-native Japanese speakers with different linguistic backgrounds and to measure the impact and use of a particular onomatopoeic expression on the perception of food. The results of this study can be sufficiently used as a part of a methodological base for education and an explanation of linguistic and semantic nuances of Japanese onomatopoeic words to non-Japanese speakers in future research. Similar experimental investigations, which were conducted among non-Japanese speakers using the onomatopoeias responsible for various nuances of pain, confirmed that these words were interpreted similarly in many dimensions without knowledge of the Japanese language [50] and the level of entropy [51] rose when mimetic words were used [52]. Thus, further experiments on food texture using mimetic words are expected to demonstrate a similar relationship between vocabulary and perception to that reported in previous studies.

While a big number of mimetic words is a specific feature of the Japanese language and their usage can be acknowledged as a limitation of this study, continuing research on sound symbolism may highlight the universal character of such words and their ability to express tiny perceptional nuances. Thus, English onomatopoeic words were used in a range of preceding studies, showing that there was a considerable effect of mimetic words usage on consumer judgements [53] of a certain product and on its palatability [19].

There is also strong evidence that English onomatopoeic words follow common patterns in sound and meaning with Arabic [54] and Persian languages [55] and, moreover, they have been used in the same applications in foods advertisements. In accordance with these results, it can be suggested that the idea of expressing palatability with onomatopoeic words could be extended to other languages including English, thus, helping us discover new linguistic ways of evaluating palatability.

Hanada reported that semantic images for food perception occurred in participants when they were presented with pictures of food without the real sensation of the food stimuli [18]. Our study revealed that verbal expressions are also responsible for arousing sensations. Thus, language-sensation correlations for food perception should be a subject for further research, as these results could improve our understanding concerning the level of verbal expression influence on the sensation sphere. Based on these results, we can predict consumer’s behavior and reaction to a number of new products and develop important conclusions for marketing research and everyday life.

Another important hypothesis that requires further research is that the knowledge of a certain vocabulary can stimulate the level of perception [16,17]. Further studies are required to explore the potential connection between vocabulary and sensations of taste and palatability. While the mimetic words used in the present study proved to be effective for expressing and evaluating perceived crispness and palatability, the extent of the universality of these words across different cultures and linguistic backgrounds should be evaluated. Nevertheless, this work demonstrated the importance of this topic for everyday sensations while increasing the quality of life by exploring the psychological roots of palatability.

## 5. Conclusions

To our knowledge, this study is the first to evaluate crispness by the TDS and TDL methods using Japanese onomatopoeic words as descriptors. While it has been demonstrated that onomatopoeias can enrich the description of a sensory experience and have a significant influence on gustation, their use in experimental evaluation remains insufficient. Our results indicated that Japanese onomatopoeias are associated with various crispness sensations and would help native speakers to discriminate between similar kinds of foods. PCA revealed that onomatopoeias for moisture characteristics are key descriptors of perceived crispness. According to our results, several words, such as SAKU-SAKU, provide pleasant associations and are responsible for positive evaluations of snacks. Conversely, other words, such as BETA-BETA, are associated with overly sticky and moist textures and, thus, provided an unpleasant image and negative evaluation of palatability. We observed that several onomatopoeias were more positively correlated with palatability and likeness scores compared to objective measures, such as loudness and mastication, and a predictive framework for evaluating palatability based on onomatopoeias was provided.

As this study was devoted to the nuances of the same product and specific Japanese expressions were used in the experiments, we recognize the limitations of the research. Especially, it remains unclear whether the association between described patterns of verbal expressions for sensory characteristics is present among non-Japanese participants. Further studies are required to investigate the impact of onomatopoeic words on the perception of foods on a broader level.

## Figures and Tables

**Figure 1 foods-10-01724-f001:**
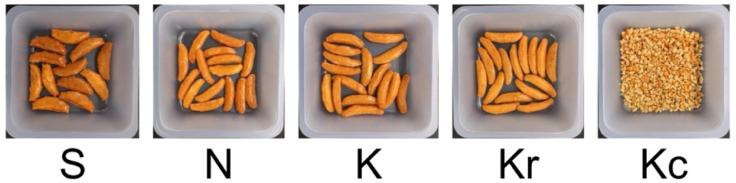
Examples of each “Kakinotane” sample. Numbers S, K, and N are available commercially and produced by different manufactures. Sample Kr was a moisturized and re-dried version of K, and Kc was a crushed version of K.

**Figure 2 foods-10-01724-f002:**
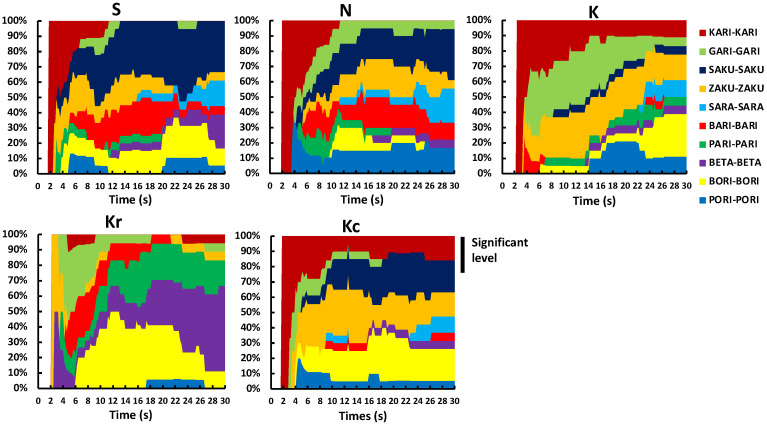
TDS band plots for each sample in Experiment 1. Samples S, K, and N are available commercially and produced by different manufactures. Sample Kr was a moisturized and re-dried version of K, and Kc was a crushed version of K. TDS, temporal dominance of sensations.

**Figure 3 foods-10-01724-f003:**
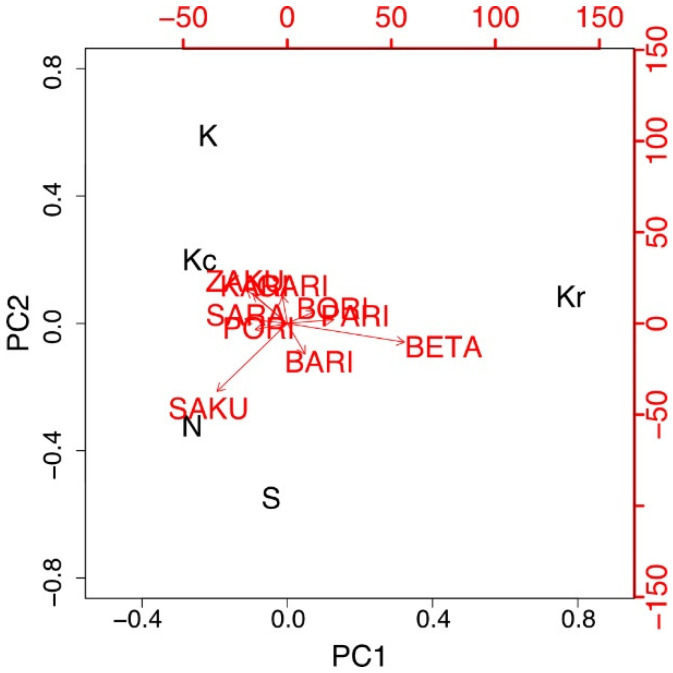
PCA biplots of the cumulative dominance duration of each mimetic word in Experiment 1. The variance in PC1 and PC2 contributed to 52% and 32%, respectively. Samples S, K, and N are available commercially and produced by different manufactures. Sample Kr was a moisturized and re-dried version of K, and Kc was a crushed version of K. PCA, principal component analysis.

**Figure 4 foods-10-01724-f004:**
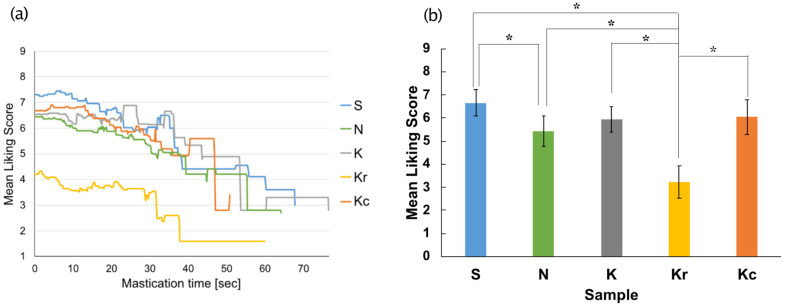
The results of the liking session in Experiment 1. (**a**) Temporal changes in the liking score for each sample. (**b**) Average liking score for each sample. ** p* < 0.05. Samples S, K, and N are available commercially and produced by different manufactures. Sample Kc was a crushed version of K, and Kr was a moisturized and re-dried version of K.

**Figure 5 foods-10-01724-f005:**
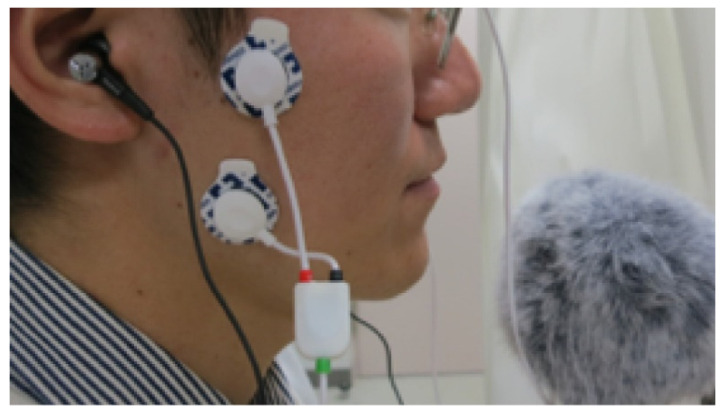
Experimental setup of the EMG probes, binaural microphones, and sound recorder with a windproof bore. EMG, electromyography.

**Figure 6 foods-10-01724-f006:**
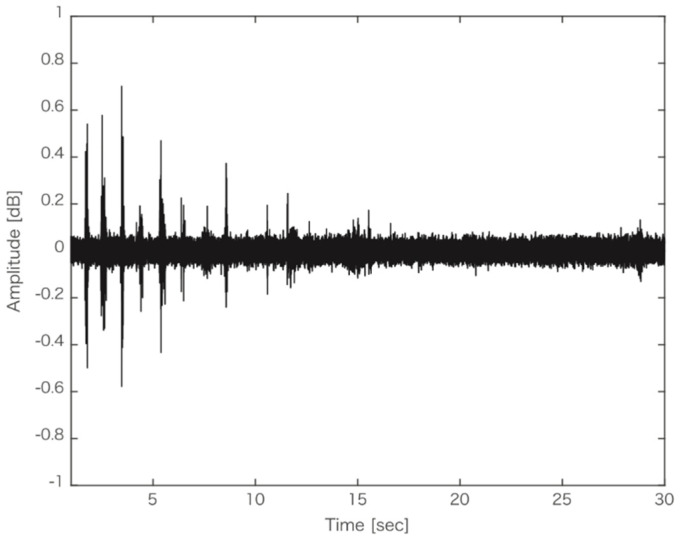
An example of the waveform of the mastication sound.

**Figure 7 foods-10-01724-f007:**
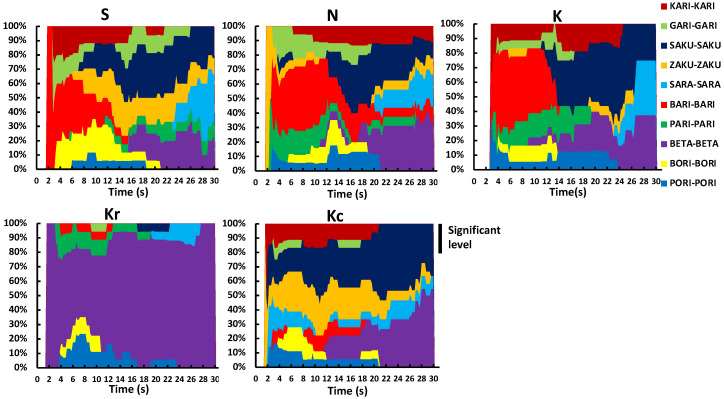
TDS band plots for each sample in Experiment 2. Samples S, K, and N are available commercially and produced by different manufactures. Sample Kr was a moisturized and re-dried version of K, and Kc was a crushed version of K. TDS, temporal dominance of sensations.

**Figure 8 foods-10-01724-f008:**
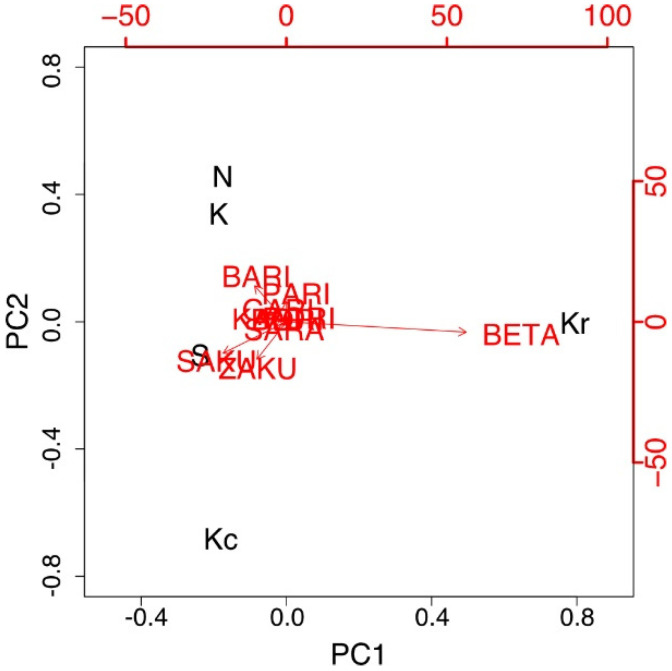
PCA biplots of the cumulative dominance duration of each mimetic word in Experiment 2. The variance of PC1 and PC2 had a contribution of 86% and 12%, respectively. Samples S, K, and N are available commercially and produced by different manufactures. Sample Kr was a moisturized and re-dried version of K, and Kc was a crushed version of K. PCA, principal component analysis.

**Figure 9 foods-10-01724-f009:**
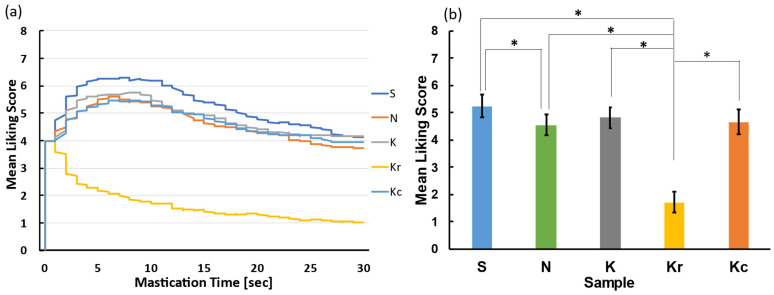
The results of the liking session in Experiment 2. (**a**) Temporal changes in the liking score for each sample. (**b**) Average liking score for each sample. * *p* < 0.05. Samples S, K, and N are available commercially and produced by different manufactures. Sample Kr was a moisturized and re-dried version of K, and Kc was a crushed version of K.

**Figure 10 foods-10-01724-f010:**
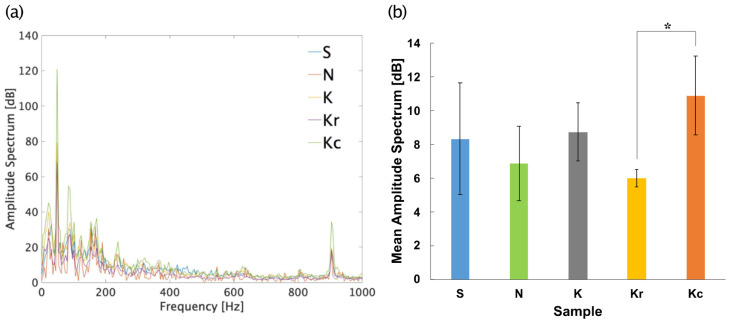
Results of the sound recording session in Experiment 2. (**a**) Power spectrum of the mastication sounds. (**b**) Mean power spectrum in each trial for each sample. * *p* < 0.05. Samples S, K, and N are available commercially and produced by different manufactures. Sample Kr was a moisturized and re-dried version of K, and Kc was a crushed version of K.

**Figure 11 foods-10-01724-f011:**
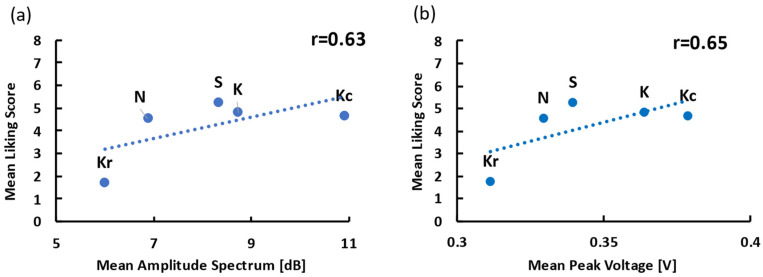
The relationships between the physiological index and the liking score. (**a**) Scatterplot of the mean amplitude spectrum and the mean liking score. (**b**) Scatterplot of the mean peak voltage and the mean liking score. Samples S, K, and N are available commercially and produced by different manufactures. Sample Kr was a moisturized and re-dried version of K, and Kc was a crushed version of K.

**Figure 12 foods-10-01724-f012:**
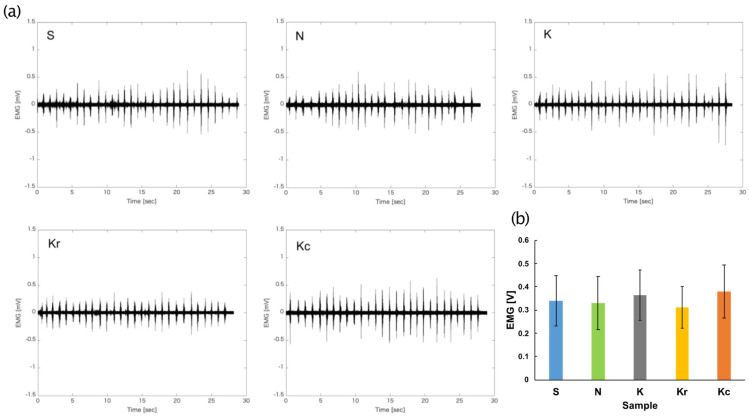
Results of the EMG recording session in Experiment 2. (**a**) Examples from a single participant during the mastication of the samples. (**b**) Mean peak voltage of the EMG of the first mastication in each trial for each sample. Samples S, K, and N are available commercially and produced by different manufactures. Sample Kr was a moisturized and re-dried version of K, and Kc was a crushed version of K. EMG, electromyography.

**Table 1 foods-10-01724-t001:** Temporal drivers of liking in Experiment 1. Samples S, K, and N are available commercially and produced by different manufactures. Sample Kr was a moisturized and re-dried version of K, and Kc was a crushed version of K. The numbers in parentheses indicate the numbers of participants who chose the corresponding descriptor during product evaluation.

	S	N	K	Kr	Kc
KARI-KARI	0.194 (2)	0.182 (2)	0.071 (2)	−0.076 (1)	0.178 (3)
GARI-GARI	-	−0.031 (1)	−0.009 (3)	0.198 (2)	0.075 (1)
SAKU-SAKU	−0.177 (3)	−0.101 (1)	-	-	0.163 (3)
ZAKU-ZAKU	0.171 (1)	0.029 (2)	−0.215 (4)	-	−0.179 (2)
SARA-SARA	-	−0.139 (1)	-	-	−0.140 (1)
BARI-BARI	0.122 (3)	0.181 (2)	-	-	-
PARI-PARI	0.015 (1)	0.074 (2)	−0.146 (1)	-	-
BETA-BETA	−0.094 (4)		-	−0.190 (1)	-
BORI-BORI	−0.001 (3)	−0.013 (2)	−0.048 (2)	0.068 (2)	0.047 (3)
PORI-PORI	−0.231 (3)	−0.182 (2)	0.347 (2)	-	−0.145 (2)

CLWD, average of individual centered liking while dominant scores. There were no statistically significant temporal drivers of liking.

**Table 2 foods-10-01724-t002:** Temporal drivers of liking in Experiment 2. Samples S, K, and N are available commercially and produced by different manufactures. Sample Kr was a moisturized and re-dried version of K, and Kc was a crushed version of K. The numbers in parentheses indicate the number of participants that cited the corresponding descriptor during product evaluation.

	S	N	K	Kr	Kc
KARI-KARI	0.045 (3)	0.015 (2)	−0.063 (3)	-	−0.012 (1)
GARI-GARI	0.041 (6)	0.056 (4)	−0.045 (2)	-	0.069 (2)
SAKU-SAKU	−0.091 (10)	−0.118 (6)	−0.045 (6)	-	0.091 (11)
ZAKU-ZAKU	0.107 (4)	−0.049 (1)	0.023 (2)	-	0.169 (9)
SARA-SARA	−0.209 (5)	−0.161 (4)	−0.248 * (4)	−0.025 (1)	−0.072 (5)
BARI-BARI	0.208 * (5)	0.365 * (8)	0.406 * (10)	0.036 (2)	0.089 (3)
PARI-PARI	0.015 (2)	0.057 (4)	0.067 (4)	0.062 (2)	-
BETA-BETA	−0.257 * (6)	−0.328 * (7)	−0.253 * (7)	−0.128 * (9)	−0.444 * (8)
BORI-BORI	0.117 * (5)	0.117 (3)	0.079 (2)	0.050 (3)	0.091 (4)
PORI-PORI	0.024 (3)	0.047 (3)	0.079 (4)	0.006 (4)	0.019 (3)

CLWD, average of individual centered liking while dominant scores. * *p* < 0.05.

## Data Availability

The data presented in this study are available on request from the corresponding author.

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
