# Peer review of "Crispness, the Key for the Palatability of “Kakinotane”: A Sensory Study with Onomatopoeic Words"

_foods, 2021, doi:10.3390/foods10081724_

Round 1
Reviewer 1 Report
The study by Saita and colleagues was aimed at investigating the association between the perceived crispness and acceptability of five types of Japanese rice crackers through TDS and TDL.
The paper is well written and the study well executed. The focus on onomatopoeic words is also interesting, however, the findings of the present study are not generalizable since both food product and expressions are specific for Japanese consumers.
Consequently, the reviewer think that these results could be more appropriate and of interest for the readers of a national journal.
Author Response
The study by Saita and colleagues was aimed at investigating the association between the perceived crispness and acceptability of five types of Japanese rice crackers through TDS and TDL.
The paper is well written and the study well executed. The focus on onomatopoeic words is also interesting, however, the findings of the present study are not generalizable since both food product and expressions are specific for Japanese consumers.
Consequently, the reviewer think that these results could be more appropriate and of interest for the readers of a national journal.
We would like to thank the reviewer for the comment. Nevertheless, we had two reasons for submitting our manuscript to the special issue of Foods.
First, this special issue is titled as follows: “Recent studying of human eating and drinking behaviors for novel foods/beverages.” Thus, we intended to submit this manuscript to the special issue because “Kakinotane,” is a famous Japanese snack, and many foreign tourists are buying this snack as a souvenir. Moreover, many foreign individuals may eat such snacks. To expand the sales of Kakinotane in foreign countries, we should understand its textural perception, and describe the perception with some descriptors. The minute differences found in textural perception of Kakinotane could not be well described using a standard English expression; thus, we used onomatopoeias as textural descriptors of them, because Japanese language has many kinds of onomatopoeias describing the minute differences of texture. We are now examining the effects of onomatopoeias on perception and description of novel sensations in non-Japanese speakers. Moreover, there are many preceding studies, which transmit the Japanese onomatopoeia feelings to non-Japanese languages, in the area of psycholinguistics. These reasons support our decision to submit this manuscript to the special issue.
Second, while a big number of mimetic words is a specific feature of the Japanese language and their usage can be acknowledged as a limitation of a current study, continuing research on sound symbolism may reveal the universal character of such words and their ability to express tiny perceptional nuances. In historical view, there are many studies that have adopted English onomatopoeic words as the descriptive words. These studies also showed that English onomatopoetic words can be applied in the other language cultures. Concerning the food texture, the number of the Japanese onomatopoetic words is larger than that of English. Interestingly, some works have showed that the Japanese onomatopoetic words can differentiate and describe the minute nuances of the food texture (Yoshikawa et al., 1970; Hanada, 2019). These findings suggested that the Japanese onomatopoetic words can be descriptive words of food texture for many individuals that use different languages. Moreover, this manuscript may enable the initiation of the application of Japanese onomatopoeia for the world-wide usage for describing the textural perception of the foods.
Reviewer 2 Report
The article deals with the texture of five types of Japanese rice crackers. The authors used the Temporal Dominance of Sensations (TDS) and Temporal Drivers of Liking (TDL) methods. In particular, the study was performed by evaluating crispness using Japanese onomatopoeic words as descriptors.
They use the TDS and TDL methods to study and describe crispness thanks to the onomatopoeic expressions is interesting even if is an empirical characterization.
Minor revisions are necessary.
1 The article title is too long, I suggest reducing it like:
2 I suggest removing lines from 96 to 103 because, in my opinion, the introduction is not the right place.
3 Fig 10 the standard error is very high, can you explain why?
4 The usage of only Japanese onomatopoeic words is restrictive, is it possible to extend the study to English word? It could be more interesting
Author Response
The authors would like to thank the reviewer for his/her constructive critique to improve the manuscript. We have made every effort to address the issues raised and to respond to all comments. Please, find next a detailed, point-by-point response to the reviewer's comments.
The article deals with the texture of five types of Japanese rice crackers. The authors used the Temporal Dominance of Sensations (TDS) and Temporal Drivers of Liking (TDL) methods. In particular, the study was performed by evaluating crispness using Japanese onomatopoeic words as descriptors.
They use the TDS and TDL methods to study and describe crispness thanks to the onomatopoeic expressions is interesting even if is an empirical characterization.
Minor revisions are necessary.
1 The article title is too long, I suggest reducing it like:
We would like to thank the reviewer for the suggestion. Following the reviewer’s suggestion, we have revised the title as follows:
“Crispness, the key for the palatability of “Kakinotane”: A sensory study with onomatopoeic words.”
2 I suggest removing lines from 96 to 103 because, in my opinion, the introduction is not the right place.
We would like to thank the reviewer for the comment. We agree with this comment. Therefore, we have removed these lines from the revised manuscript.
3 Fig 10 the standard error is very high, can you explain why?
We would like to thank the reviewer for commenting on the high SE value presented in Fig. 10. We have re-calculated the SE value and confirmed that it was right. The reason for the high SE value in the results of power spectrum of mastication might have been attributed to the high deviation found in the low frequency of power spectrum. In this study, we adopted the binaural microphones inserted into the ears. The binaural microphones are good tools for recording the bone-conduction sounds, which were affected by several factors (i.e., the anatomical structure of the skull, the condition of the occlusion, and the air flow in the ear). The differences in those personal conditions enlarged the SE value of the power spectrum of the mastication sounds.
It can be hypothesized that the high SE value was based on the small sample size. Nevertheless, the sample size was similar to that of several previous works. Moreover, the statistical parameters supported this issue; First, both the effect size and the statistical power were adequate, according to Cohen (1988). The effect size was ηp2= 0.145 and the "large" effect size in Cohen (1988) was reported as η2 = 0.138. The statistical power can be calculated from the effect size and the sample size (16 participants and five conditions) as follows: 1 - β = 0.83. Similarly, the sufficient power, as proposed by Cohen (1988), was calculated as follows: 1 - β = 0.8. Therefore, we believe that the sample size and experimental procedures were sufficient to ensure the statistical power.
Second, the difference in variance between the conditions was not a problem in performing the analysis of variance, as Mendoza's multi-sample sphericity test did not reject the assumption of sphericity (χ2(9) = 16.52, p = 0.06). Thus, we concluded that the high SE value was not caused by the limited sample size, but some other physiological reasons. We are planning to investigate the latter idea with some another physiological experiments in the future.
4 The usage of only Japanese onomatopoeic words is restrictive, is it possible to extend the study to English word? It could be more interesting.
We would like to thank the reviewer for the constructive comment. We believe that this is a unique and attractive point of our manuscript. Thus, we have added the following sentences to the Discussion section of the revised manuscript (Lines 581–592).
“While a big number of mimetic words is a specific feature of the Japanese language and their usage can be acknowledged as a limitation of this study, continuing research on sound symbolism may highlight the universal character of such words and their ability to express tiny perceptional nuances. Thus, English onomatopoeic words were used in a range of preceding studies, showing that there was a considerable effect of mimetic words usage on consumer judgements [54] of a certain product and on its palatability [20].
There is also a strong evidence that English onomatopoeic words follow common patterns in sound and meaning with Arabic [55] and Persian languages [56] and, moreover, they have been used in the same applications in foods advertisements. In accordance with these results, it can be suggested that the idea of expressing palatability with onomatopoeic words could be extended to other languages including English, thus, helping us discover new linguistic ways of evaluating palatability.”
Moreover, we have cited the following references:
Yorkston, E., Menon, G. A sound idea: phonetic effects of brand names on consumer judgements. Journal of Consumer Research. 2014., 31 (1), 43-51
Alameer, A. The linguistic features of onomatopoeia words in Arabic- English: contrastive study. International Journal of Humanities and Social Science Invention. 2019., 8 (9), 6-12
Aliyeh, K., Zeinolabedin, R. A comparison between onomatopoeia and sound symbolism in Persian and English and their application in the discourse of advertisements. International Journal of Basic Sciences & Applied Research, 3, 219-225
Reviewer 3 Report
This study investigates crispness using TDS, TDL and onomatopoeias. It is well-written and the objectives and results are clearly defined. Some minor comments are below.
Line 107- Is 21 participants enough to get meaningful results? The authors should back up the use of this number of participants by citing past studies? Also, how were the participants recruited?
Line 113- Do not need to include the three-digit codes used in the study. The authors should name the samples based on product characteristics. How were these samples chosen to be included in the study?
Line 115- Why was a crushed version included in the study?
Line 140- Did the participants place the entire sample in their mouth?
Line 414- Why is it assumed that this descriptor has two meanings?
Author Response
The authors would like to thank the reviewer for his/her constructive critique to improve the manuscript. We have made every effort to address the issues raised and to respond to all comments. Please, find next a detailed, point-by-point response to the reviewer's comments.
This study investigates crispness using TDS, TDL and onomatopoeias. It is well-written and the objectives and results are clearly defined. Some minor comments are below.
Line 107- Is 21 participants enough to get meaningful results? The authors should back up the use of this number of participants by citing past studies? Also, how were the participants recruited?
We would like to thank the reviewer for the comment. In these studies, the participants were students, graduate students and researchers who had enough experiences in experimental psychology of sensation and/or sensory studies of foods. Thus, these participants, who were not naïve consumers, had high sensory abilities required for this study. Moreover, previous studies included a similar number of participants (< 20 participants). Especially, 10, 20, and 20 participants were included in the studies of Kohyama et al. (2007), Hutchings et al. (2014), and Zampini and Spence (2004), respectively.
Please note that we have added the following part to the revised manuscript (Lines 105–108):
“The sample size of the study was determined with the reference of those in the preceding studies [10, 36]. All the participants were students, graduate students, and researchers of our department, and had enough experience in experimental psychology of sensation and/or sensory studies of foods.”
Line 113- Do not need to include the three-digit codes used in the study. The authors should name the samples based on product characteristics. How were these samples chosen to be included in the study?
We would like to thank the reviewer for the question. Three of the samples (155, 367, and 881) were produced by representative brands of “Kakinotane” and chosen based on the varieties of textural characteristics by the sensory panel in the Japanese manufacturer of rice crackers. Thus, we labeled the samples as S, N, K, Kc, and Kr instead of using the three-digit codes in this manuscript.
Please note that we have added the following part to the revised manuscript (Lines 112–114):
“Three of the samples (S, N, and K) were produced by representative brands of “Kakinotane” and chosen based on varieties of textural characteristics by the sensory panel of the Japanese manufacturer.”
Line 115- Why was a crushed version included in the study?
We would like to thank the reviewer for the question. To respond to the reviewer’s question, we have added the following sentence to the revised manuscript (Lines 116–118):
“The latest was used to investigate the difference in palatability between normal and de-structured foods, as the de-structured foods were suggested not to have the texture of the first bite.”
Line 140- Did the participants place the entire sample in their mouth?
We would like to thank the reviewer for the question. To respond to the reviewer’s question, we have added the following sentence to the revised manuscript (Lines 143–144):
“Four types of samples were presented in paper cups containing five pieces of each product (approximately 2 g).”
Line 414- Why is it assumed that this descriptor has two meanings?
We would like to thank the reviewer for the question. Please note that the onomatopoeic word BETA-BETA is used to express the nuance of texture, such as soft and moist, but also includes sticky texture, such as the Japanese rice cake.
Round 2
Reviewer 1 Report
The authors have modified the manuscript implementing paper's quality. They also have provided sufficient explanation about why this work is suitable for Foods.